# The placental vasculature is affected by changes in gene expression and glycogen-rich cells in a diet-induced obesity mouse model

**Hui Zhao***, Ronald J. Wong, David K. Stevenson

Department of Pediatrics, Division of Neonatal and Developmental Medicine, Stanford University School of Medicine, Stanford, CA, United States of America

* huizhao2@stanford.edu

**Data Availability Statement:** All relevant data are within the paper and its Supporting Information files.

## Abstract

Maternal obesity is a risk factor for pregnancy complications. Obesity caused by a high-fat diet (HFD) may alter maternal glucose/glycogen metabolism. Here, our objective was to investigate whether the placental vasculature is altered via changes in gene expression and glycogen-rich cells using a preclinical mouse model of diet-induced obesity. We subjected female FVB/N mice to one of three feeding regimens: regular chow (RC) given at preconception and during pregnancy (Control); RC given at preconception and then a HFD during pregnancy (HFD-P); or HFD initiated 4 weeks preconception and during pregnancy (HFD-PreCP). Daily food consumption and weekly maternal weights were recorded. Maternal blood glucose levels were measured at preconception and 4 gestational epochs (E6.5–E9.5, E10.5–E12.5, E13.5–E15.5, E16.5–E19.5). At E8.5–E16.5, total RNA in placentas were isolated for gene expression analyses. Placentas were also collected for HE and periodic acid Schiff's (PAS) staining and glycogen content assays. Dams in the HFD-P and HFD-PreCP groups gained significantly more weight than controls. Pre- and antenatal glucose levels were also significantly higher (15%–30%) in HFD-PreCP dams. Expression of several placental genes were also altered in HFD dams compared with controls. Consumption of the HFD also led to phenotypic and morphologic changes in glycogen trophoblasts (GlyTs) and uterine natural killer (uNK) cells. Alterations in vascularity were also observed in the labyrinth of HFD-PreCP placentas, which correlated with decreased placental efficiency. Overall, we observed that a HFD induces gestational obesity in mice, alters expression of placental genes, affects glucose homeostasis, and alters glycogen-positive GlyTs and uNK cells. All these changes may lead to impaired placental vascular development, and thus heighten the risk for pregnancy complications.

## Introduction

Being obese (body mass index $\geq 30 \, \text{kg/m}^2$) or overweight is common (over 60%) among woman at conception in the United States [1]. It is often associated with a state of low-grade chronic inflammation with a multifactorial etiology and may have a hereditary component in

**Funding:** This research was funded by the Prematurity Research Fund (DKS); the March of Dimes Prematurity Research Center at Stanford University (22-FY-169-126, DKS); the Charles B. and Ann L. Johnson Research Fund (DKS); the Christopher Hess Research Fund (DKS); the Providence Foundation Research Fund (DKS); the Roberts Foundation Research Fund (DKS); the Stanford Maternal and Child Health Research Institute (DKS); and the Bill and Melinda Gates Foundation (INV-006525, DKS). The funders had no role in study design, data collection and analysis, decision to publish, or preparation of the manuscript.

**Competing interests:** The authors have declared that no competing interests exist.

some individuals, but can also be exacerbated by a long-term consumption of an obesogenic diet [2–4]. Moreover, maternal obesity is associated with a high risk for developing several pregnancy complications, such as spontaneous miscarriages, gestational hypertension, pre-eclampsia, gestational diabetes, and preterm birth [5, 6]. Women who are obese prior to conception have an even greater risk [7]. In addition, offspring of obese women can be born large-for-gestational age (LGA) or diagnosed with intrauterine growth restriction (IUGR) [8], both of which may be dependent on the timing of the onset of hyperglycemia. Furthermore, offspring are at risk for developing obesity, cardiovascular disease, and diabetes later in adulthood as a result of fetal programming in a sex- as well as an age-specific manner [9–12], although this is still controversial with some reports showing a larger impact on female [13, 14] and others on male [15] offspring.

The root causes for these pregnancy disorders could be related to placental abnormalities. Excessive weight gain or consumption of an obesogenic diet during pregnancy not only can induce systemic metabolic changes, excessive adipose deposition, and/or aberrant growth factor and hormone secretion, but also may be associated with placental lipo-toxicity, oxidative stress, and inflammation [16–18]. It has been reported that consuming an obesogenic diet or excessive nutrients can alter expression of placental genes associated with angiogenesis and inflammation, cellular growth, and response to stress in mice [19], increase risk of placental infarction in non-human primates [20], and reduce placental vascularity in sheep [21]. Maternal obesity is also linked to an altered placental fatty acid uptake and subsequently affecting fatty acid delivery to the fetus [22].

Consumption of an obesogenic diet can also disturb placental glucose and glycogen homeostasis, which is critical for fetal growth. Glycogen, a large multibranched polysaccharide, serves as the main storage form of glucose [23]. Although it is commonly stored in the liver and skeletal muscle, glycogen is also found abundantly in the placenta. In mice, glycogen represents ~1.5% of the entire placental weight at its peak level, comparable with the mouse liver (2–4%) and skeletal muscle (0.2%) [24, 25]. Moreover, consuming a high-fat diet (HFD) can induce hyperglycemia and affect glucose tolerance in experimental mice [26–29].

Glycogen is primarily stored in two mouse placental cell populations: glycogen trophoblasts (GlyTs), which reside primarily in junctional zone and decidua regions or the fetal side of the placenta, and uterine natural killer (uNK) cells, which reside in decidua and metrial gland (MG) regions or the maternal side of the placenta [30]. In humans, extravillous trophoblasts (EVTs), which can invade into the spiral arteries to replace the endothelium, also contain glycogen. EVTs proximal to the villi contain low glycogen, while those located distally are glycogen rich [31]. Although the role of glycogen in these cells is unclear, the fact that both EVTs and uNK cells are involved in spiral artery remodeling has led us to speculate that glycogen may have other functions besides its commonly accepted role as an energy source for placental and fetal development [32, 33]. Because we have previously shown that pregnant mice with a partial deficiency of *Hmox1* have placentas with vascular defects [34, 35] and also have increased weight gain and visceral fat, we speculate that there is an association between obesity, placental vascular defects, and *Hmox1* expression.

Consumption of a HFD to induce high adiposity and obesity is a preclinical mouse model that has been widely used in several strains because of its ease of use, widespread application, and known phenotype [26–29]. In this study, we investigated if obesity affects placental vascular development and hypothesized that consumption of a HFD perturbs placental gene expression and leads to changes in placental cell populations, especially glycogen-rich cells, which can in turn induce placental vascular defects.

## Materials and methods

### Animal breeding and tissue collection

Male and female FVB/N mice (strain code 207, Charles River Laboratories, Wilmington, MA, USA) were mated at 7–10 weeks of age. All mice were housed in individually ventilated (IVC) cages (Innovive, San Diego, CA, USA) in a 25 ± 1° C room with a 12-h light:12-h dark cycle starting at 0700 and allowed free access to a normal chow (Teklad Rodent Diet, Cat. No. 2018, Envigo, Hayward, CA, USA) and pre-filled acidified drinking water (Aquavive®, Innovive). Pregnancies were confirmed by the presence of a vaginal plug at embryonic day (E)0.5. Animals were euthanized by carbon dioxide ($CO_2$) inhalation. No procedures caused any animal suffering and therefore the use of anesthesia and/or analgesia was not warranted. All studies were approved by Stanford University's Institutional Animal Care and Use Committee (protocol #14525) and conducted in adherence to the National Institutes of Health Guidelines on the Use of Laboratory Animals.

### Diets

Three rodent diets were used: a regular chow (RC, 18% fat, 58% carbohydrate, 24% protein, Envigo); a low-fat/high-carbohydrate diet (LFD, 10 kcal% fat, 70% carbohydrate, 20% protein, Cat. No. D12450B, Research Diets, Inc., New Brunswick, NJ, USA); and a very HF/low carbohydrate diet (60 kcal% fat, 20% carbohydrate, 20% protein, Cat. No. D12492, Research Diets, Inc.).

### Gene expression analyses

Total RNA was extracted using a TRIzol Reagent (Cat. No. 15596–026, Thermo Scientific, Waltham, MA, USA) from placentas collected at different gestational ages. cDNA was synthesized using a RT First Strand Kit (Cat. No. 330404, Qiagen, Redwood City, CA, USA). A PCR panel of 30 primers (listed on S1 Table) [36] was then used and RT-PCR performed using a RT$^2$-SYBR Green ROX qPCR Master Mix (Cat. No. 330524, Qiagen) on a Stratagene Mx3005P QPCR system (Agilent Technologies, Palo Alto, CA, USA). *Actb* and *Gapdh* were used as housekeeping genes. Data were analyzed using ΔΔCt and normalized to *Actb* and *Gapdh*. Fold changes in gene expression over E8.5 control levels were then calculated.

### Blood glucose measurements

4–6 hours after fasting, mice were sacrificed by $CO_2$ asphyxiation, and blood was immediately collected by intracardiac puncture using a 1-mL syringe fitted with a 28-ga needle. After discarding the first 2 drops of blood, 2–3 measurements for each sample were performed using a glucometer (OhCare Lite, ijDream Corporation, Mill Creek, WA, USA) and test strips (OhCare Lite, ijDream Corporation) and the mean ± SD calculated.

### Hematoxylin and eosin (HE) staining

Placentas were harvested at various gestational ages, immediately placed in PROTOCOL™ 10% (v/v) buffered formalin (Cat. No. 23–305510, Fisher Scientific, Pittsburgh, PA, USA) for 24 h for fixation, and then embedded in paraffin according to standard protocols. Six-micron thick tissues were sectioned using a microtome, deparaffinized, and then stained by HE (Cat. No. KTHNEPT, American MasterTech Scientific, Inc., Lodi, CA, USA). Placental morphology was visualized by light microscopy (Carl Zeiss Microimaging, Inc., Dublin, CA, USA).

## PAS staining

GlyTs [37] and uNK cells [38] were identified by the presence of glycogen using periodic acid Schiff (PAS) staining. Briefly, 5-µm sections of placental tissue were deparaffinized and hydrated with dH$_2$O. Slides were then immersed in periodic acid solution for 5 min followed by the Schiff reagent for 15 min at room temperature (Periodic Acid-Schiff Staining System, Cat. No. 395B, Sigma-Aldrich, St. Louis, MO, USA). Slides were then counterstained with hematoxylin solution for 30 sec. PAS-positive cells were identified by their purple/pink color with blue nuclei using an All-in-One BZ-X710 Fluorescence Microscope (Keyence, IL, USA).

## Isolectin B4 staining

After deparaffinization and rehydration, slides were antigen-retrieved using the IHC Antigen Retrieval solution (Cat. No. 50-187-82, Invitrogen Corporation, Carlsbad, CA, USA) and further treated with hydrogen peroxide. Biotinylated isolectin B4 (1:150 dilution, Cat. No. NC9918852, Vector Laboratories, Burlingame, CA, USA) was added. After 1 hour, slides were rinsed with phosphate buffered saline, incubated with Vectastain RTU Elite ABC reagent (Cat. No. NC9345440, Vector Laboratories) for 30 min, and then followed by the addition of 3,3′-diaminobenzidine (Cat. No. K5007, EnVision™ System, DAKO, Palo Alto, CA, USA). Isolectin B4-positive regions were identified by their brown color under light microscopy (Carl Zeiss Microimaging, Inc.).

## Glycogen content measurements

In the mouse placenta, there are histologically distinct layers: a layered placental disc (comprised from deep to superficial of the chorionic plate, labyrinth, and junctional zone); membranes that envelop the embryo (amnion and yolk sac); and the maternal region is the cap of the decidua basalis, which contains the MG, that rests on top of the embryo-derived placental disc [39]. The labyrinth contains the fetoplacental interface where nutrients, gases, and water are exchanged. The junctional zone consists of spongiotrophoblasts (SpTs) and GlyTs, both whose functions are not fully understood. The decidua is mainly composed of maternal cells (i.e. stromal, immune), but also of migrated fetal trophoblasts [40, 41]. The MG is located between myometrial muscle layers and mainly consists of infiltrated maternal immune cells, such as uNK cells and macrophages [39].

Therefore, for these measurements, placentas were sectioned into fetal placentas consisting of the labyrinth and junctional zone (F-placentas) and the MG region (representing the maternal placenta) in order to compare glycogen content in these regions. Tissues were then placed in 1.5-mL cryotubes and snap frozen in liquid nitrogen until analyses. Prior to measurements, frozen tissues were weighed, homogenized in molecular biology grade water in a ratio of 1:10 and then sonicated. After 10 min of boiling, tubes were centrifuged at high-speed (18,000g for 10 min at 4° C) to remove cellular debris. Glycogen content was then measured in 5 µL of supernatant using a Glycogen Assay kit (Cat. No. AB65620, Abcam, Boston, MA, USA) following manufacturer's instructions. Glycogen content was measured at 570 nm using a microplate reader (VersaMax™, Molecular Devices, San Jose, CA, USA) and expressed as mg per gram (mg/g) placental tissue weight.

## Statistical analyses

All data are presented as mean ± SD. For comparisons of multiple experimental groups, one-way analysis of variance (ANOVA) tests followed by Bonferroni corrections were used. Paired Student's t-tests were used for comparisons between two experimental groups. Differences

were deemed significant when $p < 0.05$. All statistics were calculated using GraphPad Prism 7.0 (San Diego, CA, USA).

## Results

### Effect of diet on non-pregnant mice

4-week-old non-pregnant female mice were given the RC, LFD, or HFD for a period of 4 weeks. Daily food intake and weekly body weights were recorded. Dams, who were fed the HFD, consumed ~28% more chow/day (4.2 ± 0.5 g) than those fed the LFD (2.7 ± 0.1 g) or RC (3.0 ± 0.4 g). Within 2 weeks after starting the HFD, mice gained significantly more weight compared with those fed the RC or LFD (Fig 1A). After 4 weeks, mice gained 12.5 ± 2.0 g (n = 5), a significant increase of 2.1- and 1.7-fold over those fed the LFD (5.9 ± 2.0 g, n = 5, $p < 0.001$) and RC (7.3 ± 1.4 g, n = 5, $p < 0.001$), respectively. Weight gains however were no different between the LFD- and RC-fed groups at any timepoint.

After 4 weeks, blood glucose levels in non-pregnant mice were measured 4–6 h post-fasting to avoid any possible adverse effects of extensive fasting (> 8–12 h). Blood was collected by intracardiac puncture immediately following euthanasia by $CO_2$ asphyxiation. Mice fed the HFD (n = 5) had significantly higher glucose levels (387 ± 17 mg/dL) compared with those fed the LFD (327 ± 30 mg/dL, n = 5, $p < 0.005$) or RC (323 ± 16 mg/dL, n = 4, $p < 0.005$) (Fig 1B). In addition, glucose levels were not significantly different between mice fed the LFD or RC.

### Effect of the HFD or RC during pregnancy

Since non-pregnant mice fed the LFD or RC showed no significant differences in weight gain or glucose levels, the RC was used as the control diet for the remainder of our experiments.

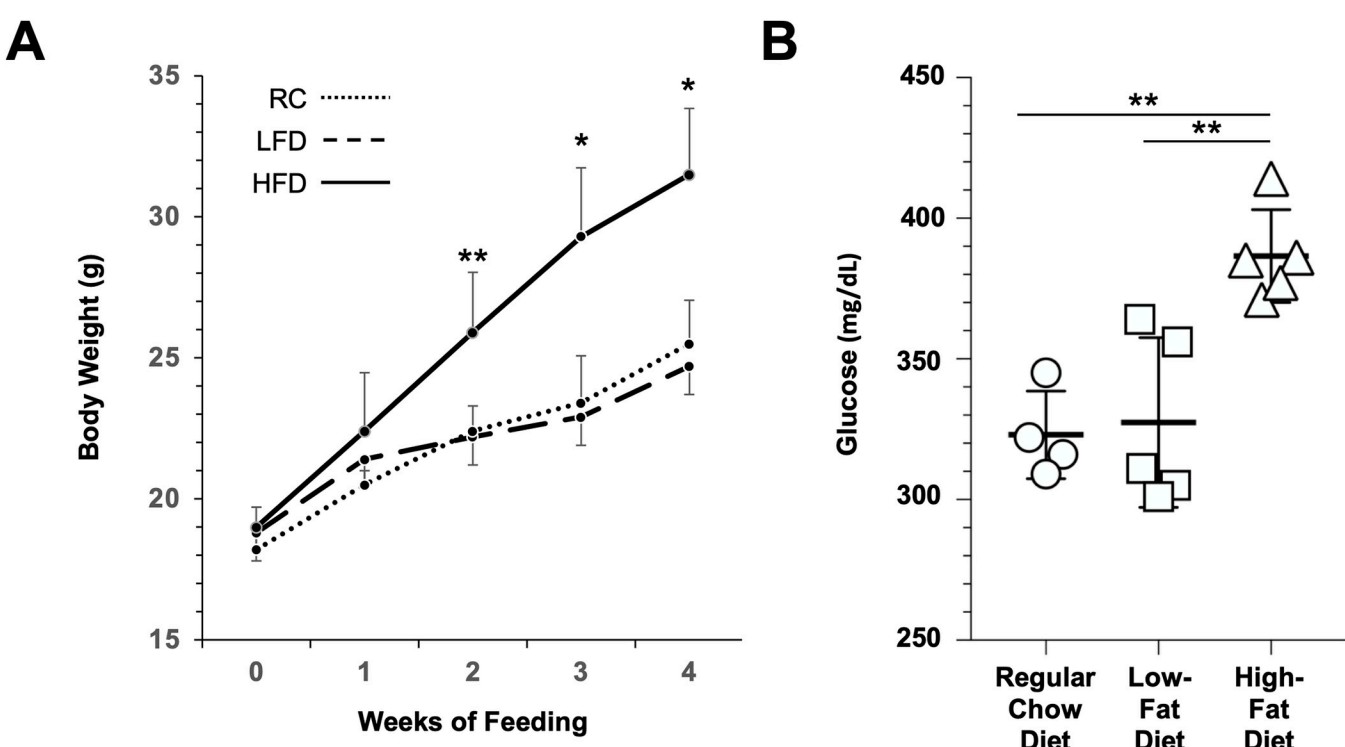

**Fig 1. Effects of different diets on non-pregnant mice.** (A) Body weights of female mice fed the RC (dotted line, n = 5), LFD (dashed line, n = 5), or HFD (solid line, n = 5) for 4 weeks. (B) Blood glucose levels in mice fed the RC (circles, n = 4), LFD (squares, n = 5), or HFD (triangles, n = 5). Data are presented as mean ± SD. *$p < 0.001$, **$p < 0.005$ vs the RC and LFD groups using a one-way ANOVA test with Bonferroni correction.

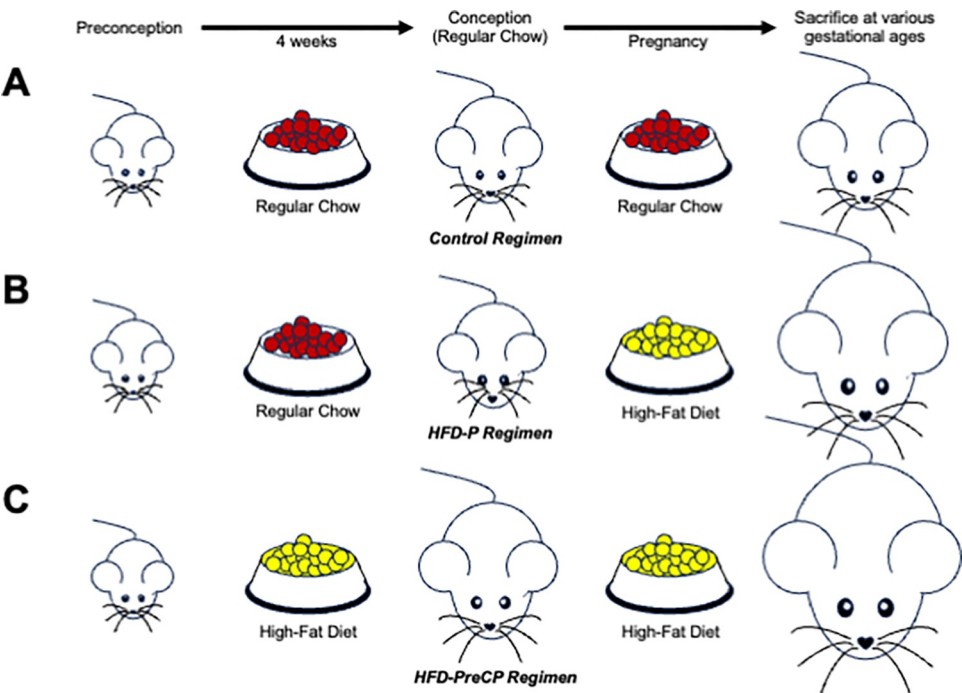

**Fig 2. Feeding regimens.** (A) Control regimen: RC at preconception and throughout pregnancy. (B) HFD-P regimen: RC at preconception and the HFD throughout pregnancy. (C) HFD-PreCP regimen: the HFD initiated 4 weeks preconception and throughout pregnancy.

In this study, female mice were given one of the following three feeding regimens: the RC given at preconception and throughout pregnancy (Control, Fig 2A); the RC given at preconception and then the HFD throughout pregnancy (HFD-P, Fig 2B); or the HFD initiated 4 weeks preconception and throughout pregnancy (HFD-PreCP, Fig 2C).

As expected, in mice receiving either the HFD-P or HFD-PreCP regimen, body weights increased throughout gestation, with HFD-PreCP mice gaining more weight compared with controls. Specifically, when compared with controls, body weights of HFD-P mice increased 15.2% (n = 10), 16.6% (n = 10), and 17.0% (n = 10), while HFD-PreCP mice increased 39.2% (n = 7), 38.6% (n = 7), and 24.3% (n = 7) at E12.5, E14.5, and E16.5, respectively (Fig 3A).

Over the course of being subjected to either the HFD-P or HFD-PreCP regimen, dams displayed "ruffled", "sweaty appearing" fur, indicative of poor health as well as central adiposity and visceral fat noted at sacrifice. The numbers of viable fetuses per dam at sacrifice were not however significantly different between HFD-P (8.8 ± 1.7 pups, n = 17 pregnancies) or HFD-PreCP (8.3 ± 1.1 pups, n = 19 pregnancies) groups compared with controls (9.1 ± 1.5 pups, n = 17 pregnancies).

Moreover, consumption of the HFD significantly increased fetal weight at E14.5 only in the HFD-P group (S2 Table). At E16.5, fetuses harvested from dams given the Control, HFD-P, and HFD-PreCP regimens weighed 516 ± 56 mg (n = 15, from 3 dams), 545 ± 37 mg (n = 17, from 3 dams), and 532 ± 48 mg (n = 17, from 3 dams), respectively. These data showed slight (but not significant) increases of 5.8% and 3.1% in fetal weight after mothers were given the HFD-P and HFD-PreCP feeding regimens, respectively, compared with that of control dams (Fig 3B and S2 Table). In contrast, weights were similar between fetuses harvested from HFD-P and HFD-PreCP dams.

### Effect of the HFD on maternal glucose levels during pregnancy

To investigate if consumption of the HFD could affect maternal glucose homeostasis, we measured glucose levels in dams subjected to the HFD-P and HFD-PreCP regimens at preconception as well as over 4 gestational epochs–representative of early (E6.5–E9.5), early mid- (E10.5–12.5), mid- (E13.5–E15.5), and late gestation (E16.5–E19.5). We found that mean glucose levels in the HFD-PreCP dams were significantly higher at preconception (387 ± 17 mg/dL, n = 5) and at the two early gestational epochs [E6.5–E9.5 (348 ± 39 mg/dL, n = 12) and E10.5–E12.5 (331 ± 33 mg/dL, n = 5)] compared with control levels (323 ± 16 mg/dL, n = 4; 308 ± 47 mg/dL, n = 15; 258 ± 29 mg/dL, n = 13, respectively, Fig 3C). Moreover, these differences diminished when pregnancy entered mid- and late gestational epochs. By the early midgestation epoch (E10.5–E12.5), glucose levels in the HFD-P group were comparable to those of dams in the HFD-PreCP group.

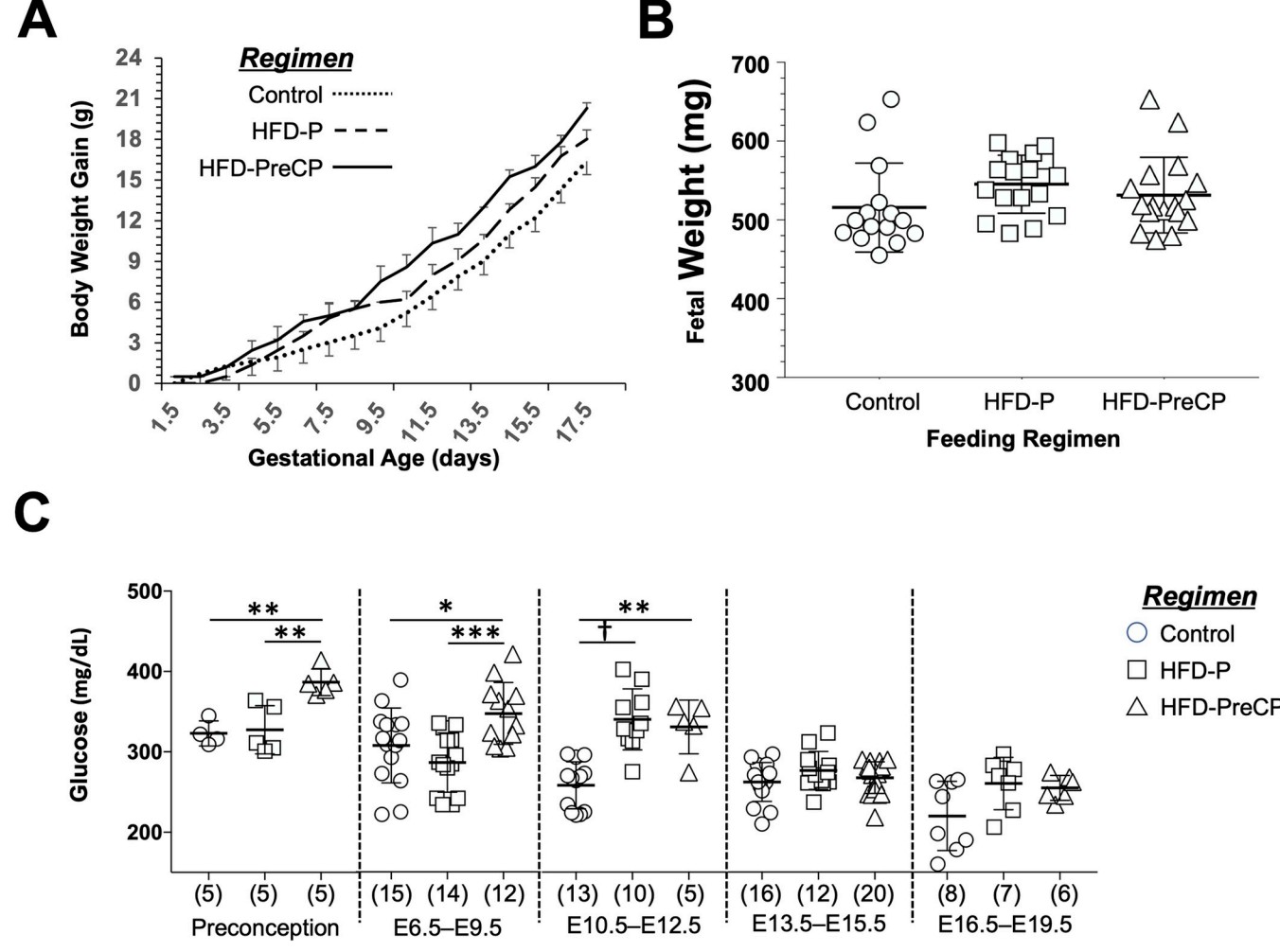

**Fig 3. Effects of diet regimens on pregnant mice.** (A) Body weight gain during pregnancy in mice fed the Control (dotted line, n = 16), HFD-P (dashed line, n = 10), or HFD-PreCP (solid line, n = 7) regimen. (B) Weights (g) of fetuses harvested at E16.5 from pregnant mice fed the Control (circles, n = 15), HFD-P (squares, n = 17), or HFD-PreCP (triangles, n = 17) regimen. C) Maternal glucose levels at 5 different gestational epochs: preconception, E6.5–E9.5; E10.5–12.5; E13.5–E15.5; and E16.5–E19.5 in dams fed the Control (circles), HFD-P (squares), or HFD-PreCP (triangles) regimens. Number of samples per group are shown in parentheses. Data are presented as mean ± SD. *p < 0.05, **p < 0.005, ***p < 0.0005, †p < 0.0001 using one-way ANOVA tests with Bonferroni corrections.

## Effect of the HFD on placental gene expression during pregnancy

To investigate if consumption of the HFD could affect placental gene expression, we first collected placentas at different gestational ages. Expression of a panel of 30 genes encoding proteins that we had previously found to be important for placental vascular development [36] was evaluated using a customized PCR panel (S1 Table). Among them were hypoxia response factors (i.e. hypoxia-inducible factor 1-alpha and 2-alpha [*Hif1a*, *Hif2a*]), antioxidant proteins (i.e. superoxide dismutase 1 and 2 [*Sod1*, *Sod2*]; glutathione peroxidase 1 and 4 [*Gpx1*, *Gpx4*], heme oxygenase-1 and -2 [*Hmox1*, *Hmox2*]), angiogenic factors (i.e. placental growth factor [*Pgf*]; stromal cell-derived factor 1 [*Cxcl12*] and its receptor [*Cxcr4*]; vascular endothelial growth factor A [*Vegfa*] and its receptors [*Vegfr1*, *Vegfr2*]); matrix metalloproteinase-9 and -2 [*Mmp9*, *Mmp2*]; angiopoietin-1 [*Angpt1*]; and transcription factors (i.e. specificity protein 1 [*Sp1*], activator protein 2 [*Ap2*], nuclear factor erythroid 2-related factor 2 [*Nrf2*], and basic leucine zipper transcription factor 1 [*Bach1*]).

At early/mid- (E9.5) and mid- (E11.5) gestational ages, the expression levels of several genes were significantly altered in HFD-P mice when normalized to E8.5 levels of controls in order to control for any age-dependent changes in expression over pregnancy. Genes encoding for *Hif1*, *Angpt1*, *Pgf*, *Hmox1*, *Sod1*, *Ap2*, and *Nrf2* were significantly downregulated, while the gene for *Gpx1* was significantly upregulated (Fig 4).

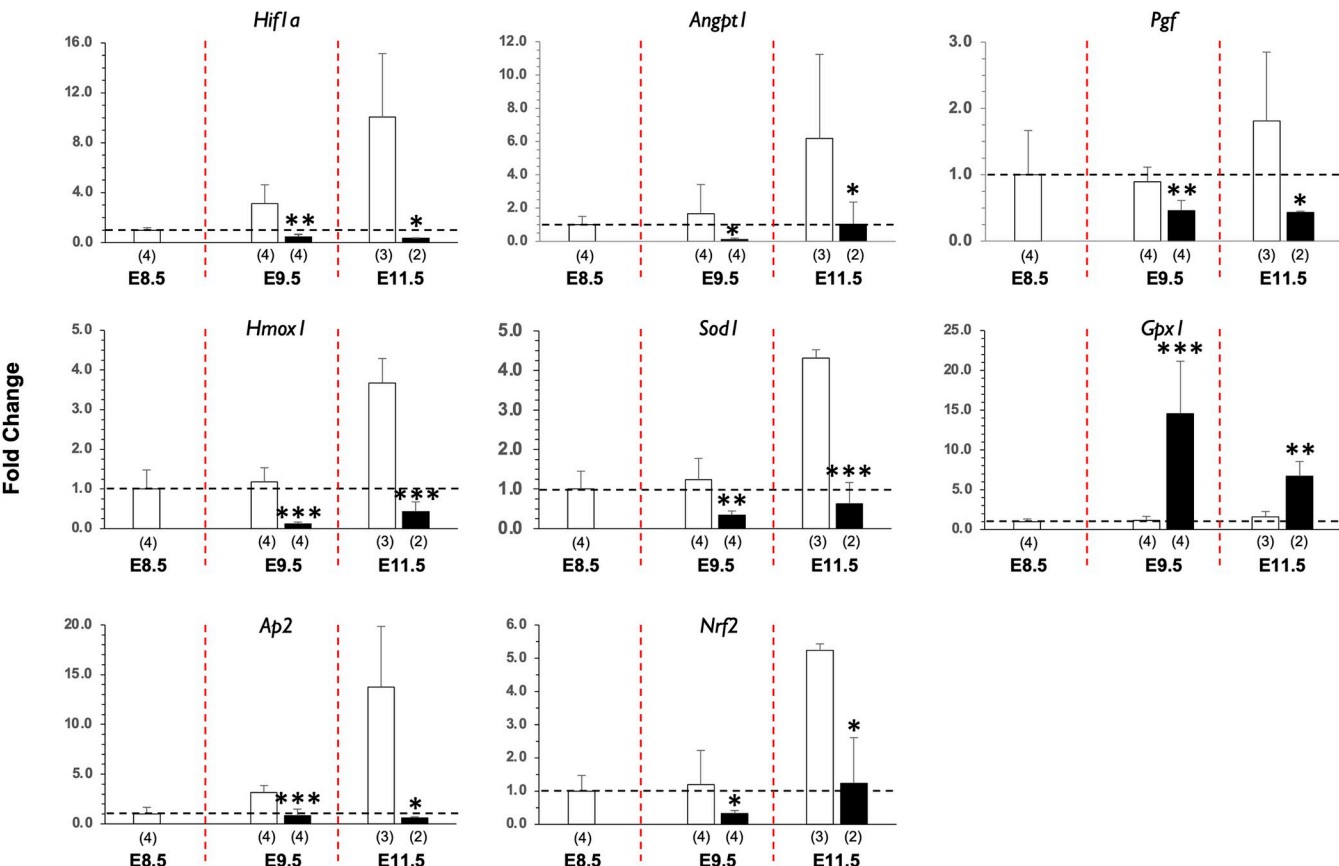

**Fig 4. Placental gene expression at E9.5 and E11.5.** Alterations in gene expression (*Hif1a*, *Angpt1*, *Pgf*, *Hmox1*, *Sod1*, *Gpx1*, *Ap2*, and *Nrf2*) as measured using a custom qRT-PCR panel were found in placentas harvested from pregnant mice subjected to the Control (white bars) or HFD-P (black bars) regimen. Data are presented as mean ± SD fold change from the E8.5 RC group results. Number of samples per gestational age epoch per group are shown in parentheses.

### Effect of the HFD on glycogen-positive GlyTs and uNK cells in the placenta

To identify and characterize glycogen-rich cells, placentas were collected at E14.5 and sectioned for PAS staining. Glycogen-positive cells were identified by their purple/pink color and blue/gray nuclei and present in all placental layers in control placentas (Fig 5). In the labyrinth, islets of PAS-positive cells were found to reside deep in the vasculature (Fig 5G and 5H). They were found clustered together and wrapped by a layer of PAS-negative cells. Cytoplasms appeared less vacuolated compared with mature GlyT cells found in the junctional zone/decidua (Fig 5E and 5F), and therefore may be GlyT precursors (Pre-GlyT). In the junctional zone/decidua, GlyT cells, identified by their characteristic vacuolated appearances, strong PAS-positive staining, and pattern of distribution, were observed (Fig 5E and 5F). GlyT cells in the junctional zone were clustered together while those that migrated into the decidua appeared more diffusely distributed, mixing in with other cell types, and sometimes without clear cell borders. Some GlyT cells were present surrounding the maternal blood canals

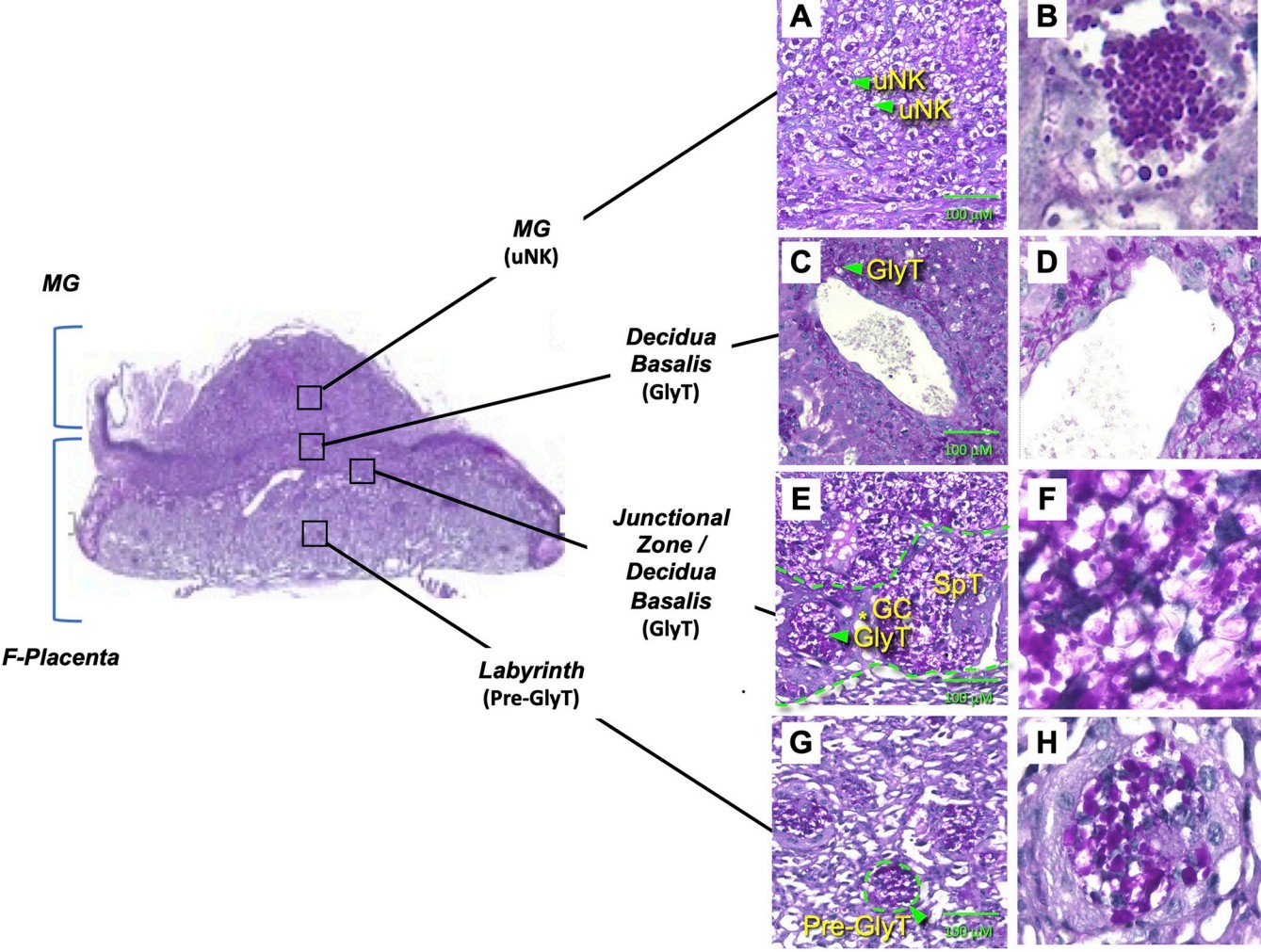

**Fig 5. Glycogen-rich GlyTs and uNK cells were identified by PAS staining in E14.5 placentas from mice fed the RC.** (A, B) uNK cells in the MG. (C, D) GlyTs surrounding the maternal blood canals in the decidua basalis region. (E, F) GlyTs in the junctional zone/decidua basalis region. (G, H) Pre-GlyT islets in the labyrinth. Junctional zone regions and cell islets in the labyrinth region are delineated by dashed green lines. Green triangles show GlyTs, SpTs, or uNK cells and the yellow asterisk denotes a giant cell (GC). n of 5 placentas from 3 pregnancies evaluated for each staining. Images on the left and right panels are shown at 200x and 400x magnification, respectively.

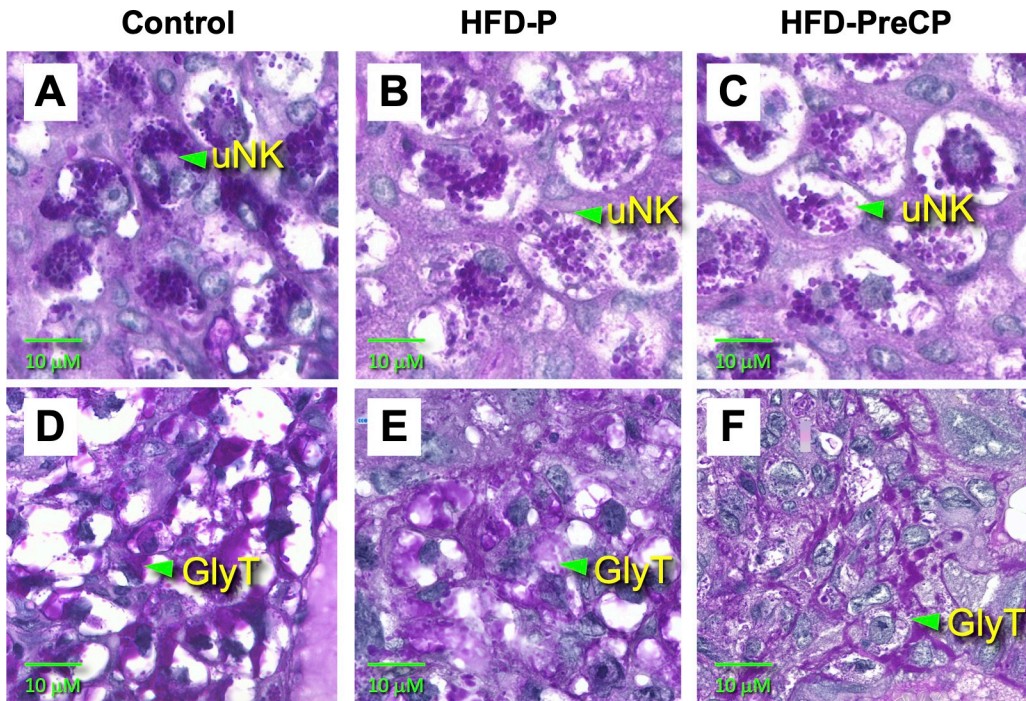

**Control**  **HFD-P**  **HFD-PreCP**

**Fig 6. PAS-positive uNK (upper panels) and GlyTs (lower panels) in placentas.** Placentas were harvested from mice fed the: (A, D) Control; (B, E) HFD-P; or (C, F) HFD-PreCP regimen. Green triangles show GlyTs or uNK cells. n = 5 placentas from 3 pregnancies evaluated for each staining. Images are shown at 400x magnification.

(Fig 5C and 5D). In the MG (Fig 5A and 5B), large vacuolated cells with many PAS-positive staining granules in the cytoplasm were observed. Based on previous reports [30, 38], we classified these types of cells as uNK cells.

Next, to investigate how consumption of the HFD and subsequent development of obesity affect the morphologies and phenotypes of glycogen-rich GlyT and uNK cells, placentas were harvested at E14.5 from each diet group and then compared. Representative images of PAS-stained E14.5 placentas are shown in Fig 6. In the junctional zone/decidua of control placentas, purple/pink color PAS staining was mostly found in the cytoplasm of vacuolated GlyT cells (Fig 6D). In HFD-PreCP placentas, however, there appeared less vacuolated GlyT cells and PAS-positive staining was found not in the cytoplasm, but on cell membranes (Fig 6F). The significance of glycogen distribution in GlyT are not well known and it might be associated with altered glucose/glycogen metabolism [42]. The changes of GlyT cells in HFD-P placentas (Fig 6E) was not as dramatic as those in HFD-PreCP placentas, but less vacuolated GlyT cells were also observed. Due to the complexity and dynamic properties of the placenta and absence of clear boundaries of GlyT cells, we were unable to accurately quantify the number of glycogen-rich cells in placentas from each diet group. In the MG, the size (15–30 μm) and morphology of uNK cells were similar among all diet groups. However, intensity of PAS-positive granules in uNK cells appeared reduced in both HFD-P and HFD-PreCP placentas (Fig 6B and 6C) compared with those of controls (Fig 6A).

## Effect of the HFD on placental vascular formation

To study if consumption of the HFD can affect the placental vasculature in the fetoplacental interface, labyrinths of placentas from the different diet regimen groups were sectioned for HE

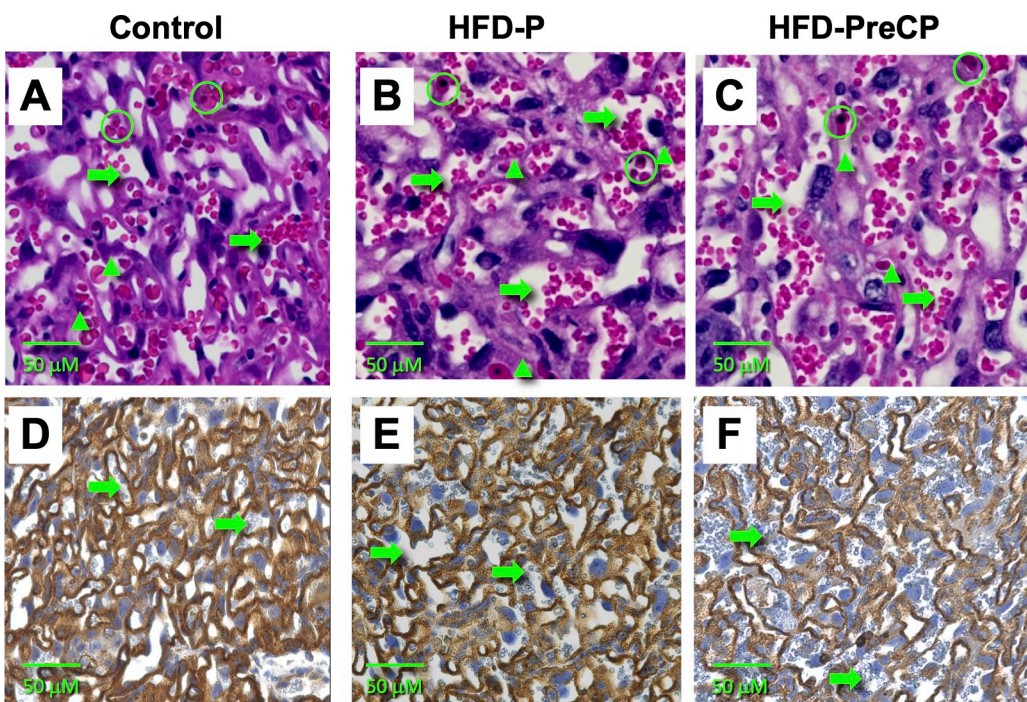

**Fig 7. HE (upper panels) or isolectin B4 (lower panels) staining of labyrinth regions.** Placentas were harvested from dams subjected to the: (A, D) Control; (B, E) HFD-P; or (C, F) HFD-PreCP regimen. Maternal red blood cells (non-nucleated) were small and shown clustered, while fetal red blood cells (nucleated) were isolated and denoted by green rings in the upper panels. Maternal sinusoid spaces are indicated by the green arrows, while fetal capillary spaces are indicated by the green triangles. n = 5 placentas from 3 pregnancies evaluated for each staining. Images are shown at 400x magnification.

staining (Fig 7A to 7C) to assess morphology and for biotinylated B4 staining (Fig 7D to 7F) to identify endothelial cells surrounding fetal blood vessels.

HE staining revealed that maternal blood spaces in the labyrinth of control placentas were relatively compact and densely formed (Fig 7A). In HFD-P placentas, large maternal sinusoids were occasionally seen (Fig 7B). However, in HFD-PreCP placentas, maternal sinusoids appeared elongated and dilated, and formed a loose-appearing structure (Fig 7C).

Isolectin B4 staining confirmed our findings above. Although B4-positive fetal endothelial layers appeared intact in placentas from all diet regimen groups, their vascular structures were different. Compared with the well-knit vascular network found in control placentas (Fig 7D), HFD-PreCP placentas had elongated fetal vessel boundaries and less dense fetal capillary networks (Fig 7F), with broader maternal spaces and areas for nutrient exchange. These structural changes in HFD-P placentas were not as apparent as those in HFD-PreCP placentas, but in some areas, there were fewer fetal capillary structures (Fig 7E). Taken together, these findings of less densely-appearing vascular branching in the labyrinth suggest a "hypo-vascularity" within HFD placentas.

## Effect of the HFD on placental efficiency (PE)

To further understand how consumption of the HFD can affect the fetomaternal interface, we separated placentas into its two distinct parts: the F-Placenta, containing the labyrinth, junctional zone, and decidua basalis; and the MG, consisting of cells derived from the uterus or have infiltrated from the maternal circulation (Fig 5).

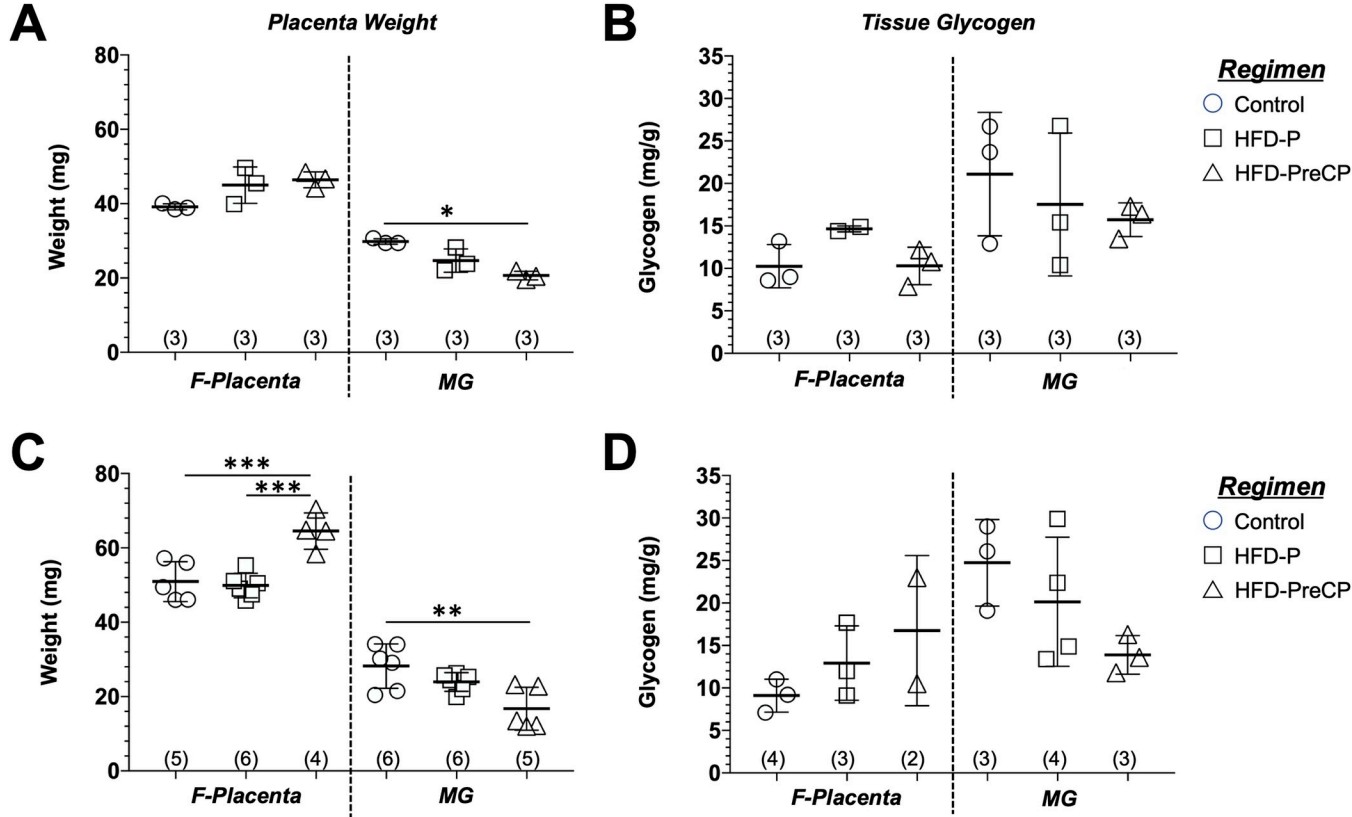

**Fig 8. Weights and glycogen content in placentas.** Placentas were harvested at (A, B) E14.5 and (C, D) E16.5 from pregnant mice subjected to the Control (circles), HFD-P (squares), or HFD-PreCP (triangles) regimens. Each placenta was then separated into the F-Placenta and MG. Weights (mg) of F-Placentas and MGs are shown at (A) E14.5 and (C) E16.5. Glycogen content is shown as mg/g placenta at (B) E14.5 and at (D) E16.5 with controls shown as circles; HFD-P as squares; and HFD-PreCP as triangles. Number of samples per group are shown in parentheses. Data are presented as mean ± SD. *p < 0.02; **p < 0.01; ***p < 0.005 using one-way ANOVA tests with Bonferroni correction.

We found that weights of F-placentas harvested at E14.5 were not different from those harvested from dams subjected to either of the three diet regimens (Fig 8A). At E16.5, HFD-PreCP F-placentas weighed significantly more than both control (p < 0.005) and HFD-Pre-CP (p < 0.005) F-placentas (Fig 8C). Interestingly, no difference was found between F-Placentas from HFD-P and control mice. For the MG, those from HFD-Pre-CP placentas weighed significantly less compared with those from controls at both E14.5 (p < 0.05, Fig 8A) and E16.5 (p < 0.01, Fig 8C). For all diet regimen groups, MGs weighed significantly less than F-Placentas at both E14.5 and E16.5.

We then calculated PE defined as the ratio of fetal (or birth) weight to placental weight, which is used commonly as a marker of placental function [43]. The mean weights of the fetus and of F-Placentas (containing fetoplacental interface) were used to calculate PE for each diet group. S2 Table shows that compared with control and HFD-P placentas, PE decreased 15% and 7%, respectively, at E14.5 and 19% and 25%, respectively, at E16.5 in HFD-PreCP placentas, suggesting a reduction in nutrient transport capacity or a failure to adapt.

### Effect of the HFD on tissue glycogen content in the MG during pregnancy

We then further compared tissue glycogen quantitatively in placentas harvested from mice subjected to the different feeding regimens. Glycogen per gram placental tissue (mg/g) in

F-Placentas and MGs were determined and compared among the different diet regimen groups at E14.5 and E16.5. For F-placentas, tissue glycogen content at E14.5 was slightly higher (1.4-fold) in those from HFD-P dams (14.6 ± 0.4 mg/g) compared with HFD-PreCP (10.3 ± 2.2 mg/g) and controls (10.3 ± 2.6 mg/g, Fig 8B). However, at E16.5, tissue glycogen contents increased in HFD-P (12.9 ± 4.4 mg/g) and HFD-PreCP (16.8 ± 8.8 mg/g) mice compared with controls (9.5 ± 1.8 mg/g) mice (Fig 8D). In contrast, for the MGs, tissue glycogen contents in HFD-P (17.5 ± 8.4 mg/g) and HFD-PreCP (15.7 ± 2.0 mg/g) mice slightly decreased at E14.5 down 17% and 25%, respectively, compared with controls (21.1 ± 7.2 mg/g, Fig 8B). At E16.5, we observed a similar trend with tissue glycogen contents in HFD-P (20.2 ± 7.6 mg/g) and HFD-PreCP (13.9 ± 2.3 mg/g) mice decreased 18% and 44%, respectively, compared with controls (24.7 ± 5.1 mg/g, Fig 8D). However, these changes were not statistically significantly different.

## Discussion

In this study, we used a HFD to induce obesity in mice to mimic the high-density food consumption in modern society. Preclinical murine models of genetic and nutrient obesity have been widely used to study obesity and diabetes [26, 28] and the disease progression has been well documented [44, 45]. The C57Bl/6J mouse is the most widely used strain in studies of obesity [46], but here we used FVB/N mice because this study is a further extension of our previous studies investigating the role of HO-1 deficiency in placental vascular development, HO-1 heterozygote transgenic mice, which was created on an FVB/N background. Importantly, Nascimento-Sales et al [47] have shown that FVB/N mice are not resistant to HFD-induced obesity as previously believed, with weight gain and worsened glucose intolerance following consumption of an HFD and thus can also be used to elucidate aspects of obesity and even beyond those established in the C57Bl/J6 strain.

As early as 3 days of consuming a HFD, pancreatic B-cell proliferation has already increased [45]. Body weight gain and moderate hyperglycemia are observed in the adaptive phase (1–7 weeks of consuming a HFD) followed by a plateau stage (8–16 weeks of consuming a HFD), but other important disease conditions, such as insulin resistance, inflammation, and tissue oxidative stress, progressively worsen [44, 45]. After 4–6 months of a HFD, diabetic symptoms, such as hyperglycemia, hypercholesterolemia, insulin resistance, and hepatic dysfunction gradually emerge [28]. In this study, we found that consumption of a HFD for 4 weeks in non-pregnant mice (Fig 1A) as well as for 2–3 weeks during pregnancy (HFD-P) or for 6–8 weeks (HFD-PreCP) (Fig 3A) was sufficient to increase maternal body weight and blood glucose levels (Figs 1B and 3C). Accordingly, our study protocol likely falls within the spectrum of the adaptive phase of disease progression with significant increases in blood glucose early in gestation (E6.5–E12.5, Fig 3C). Therefore, our goal was to investigate the effect of a HFD during pregnancy on placental vascular development. Other complications related to diabetes were not investigated and beyond the scope of these studies.

Of note, we found that consumption of a HFD could alter the expression of specific placental genes (Fig 4). After screening a panel of genes associated with placental vascular formation, we found changes in eight genes related to oxygen sensing, oxidative stress, and angiogenesis in the HFD-P and HFD-PreCP groups. The fact that consumption of a HFD could affect placental gene expression has also been reported by others [19], suggesting that diet alone can impact placental development. One critical gene that is altered is *Hif1a*, a molecular sensor of oxygen levels and global regulator for more than 70 genes, including genes for angiogenesis, oxygen supply, stemness/self-renewal, proliferation, epithelial to mesenchymal transition, metastasis and invasion, redox homeostasis, and apoptosis [48, 49]. Another important gene is

*Hmox1*, which is a stress-responsive protein with antioxidant properties. Our previous studies [34, 35] have demonstrated that placentas from pregnant mice with a partial *Hmox1* deficiency are obese and have impaired vascular development, such as a thinner junctional zone region and impaired spiral artery remodeling, suggesting that HFD-induced placental vascular defects might be associated with *Hmox1* function.

Alterations in gene expression may also negatively affect GlyTs. Tunster et al. have reviewed more than 40 mouse lines with targeted gene mutations that lead to abnormal GlyT phenotypes either with or without aberrant glycogen storage [33]. Among them are mouse strains with deficiencies in *Hif1a*, *Hif1b*, or *Pgf*. A deficiency in either *Hif1a* or *Hif1b* (*Arnt*) resulted in impaired GlyT differentiation. A mutation in *Pgf* was found to affect both glycogen differentiation and storage [33]. Interestingly, we observed that both *Hif1a* and *Pgf* were downregulated in placentas from dams fed a HFD (Fig 4) and the changes in phenotype and glycogen deposition in GlyTs in HFD-PreCP placentas (Fig 6C and 6F). The first glycogen-rich cells are present in the ectoplacental cone around E6.5 [50, 51] followed by the appearance of mature GlyTs in the junctional zone region around E12.5 [52]. GlyTs, SpTs, and trophoblast giant cells all stem from the same progenitor populations in ectoplacental cone in mice [33]. More studies are needed to be done to further our understanding on the roles of *Hif1a* and *Pgf* on GlyT lineage development.

Moreover, diet and blood glucose levels may also be associated with GlyT differentiation and glycogen storage (Fig 6). Aberrant placental glycogen storage in humans has been found to be associated with maternal diabetes and pre-eclampsia [33]. It is commonly believed that under healthy conditions, hyperglycemia induces insulin production to promote glycogen storage in the liver, skeletal muscle, and fat cells [53]. However, if insulin resistance develops during long-term consumption of an obesogenic diet, then glycogen storage is dysregulated [53]. Our data shows that consuming an obesogenic diet increases maternal glucose levels (Fig 3C), and also changes glycogen deposition and cellular distribution (Figs 5–7) in GlyT cells. It is currently not well understood how circulating glucose affects GlyT development. Insulin and insulin-like growth factors have been suggested in the receptor-mediated regulation of placental growth and transport, trophoblast invasion, and placental angiogenesis [54]. Mahany et al. have reported that in placentas of mice fed a HFD, there is significant upregulation of insulin-like growth factor 1 (*Igf1*) and insulin-like growth factor binding protein 1 (*Igfbp1*) [19], both are involved in glucose and insulin regulation. How this system affects glycogen content in GlyTs remains to be determined.

uNK cells are a unique NK cell population that produces factors that attract EVTs to promote trophoblast invasion and spiral artery remodeling [30]. Mouse uNK cells can be identified as lymphocytes containing PAS-reactive cytoplasmic granules [30]. In this study, we found decreases in mass (Fig 8C) at E16.5 as well as less signal intensity of PAS-positive granules in uNK cells (Fig 6) at E14.5 in the HFD-PreCP placentas. All these finding suggested that HFD may affect MG formation and uNK cell behavior. It remains unclear how a HFD affects glycogen content in uNK cells and its subsequent effects on their infiltration, differentiation, and function in the uterus and MG. As reported, NK cells preferentially utilize glucose, and their function and glucose metabolism are highly integrated. Aberrant NK cell function has been observed in obese patients. Their circulating NK cells exhibit reduced frequencies, diminished cytotoxicity, and impaired interferon gamma (IFNγ) production [55–57]. In addition, NK cell cytokine production can be regulated by the glycogen synthase kinase-3 (GSK3)-dependent pathway. Repression of GSK3 has been found to restore NK cell cytotoxicity in acute myeloid leukemia patients [58]. All these findings suggest that a possible crosstalk exists between NK function and cellular glucose metabolic homeostasis. Since we also observed vascular changes in fetomaternal interface (Fig 7F) and a lower PE (S2 Table) in HFD-PreCP

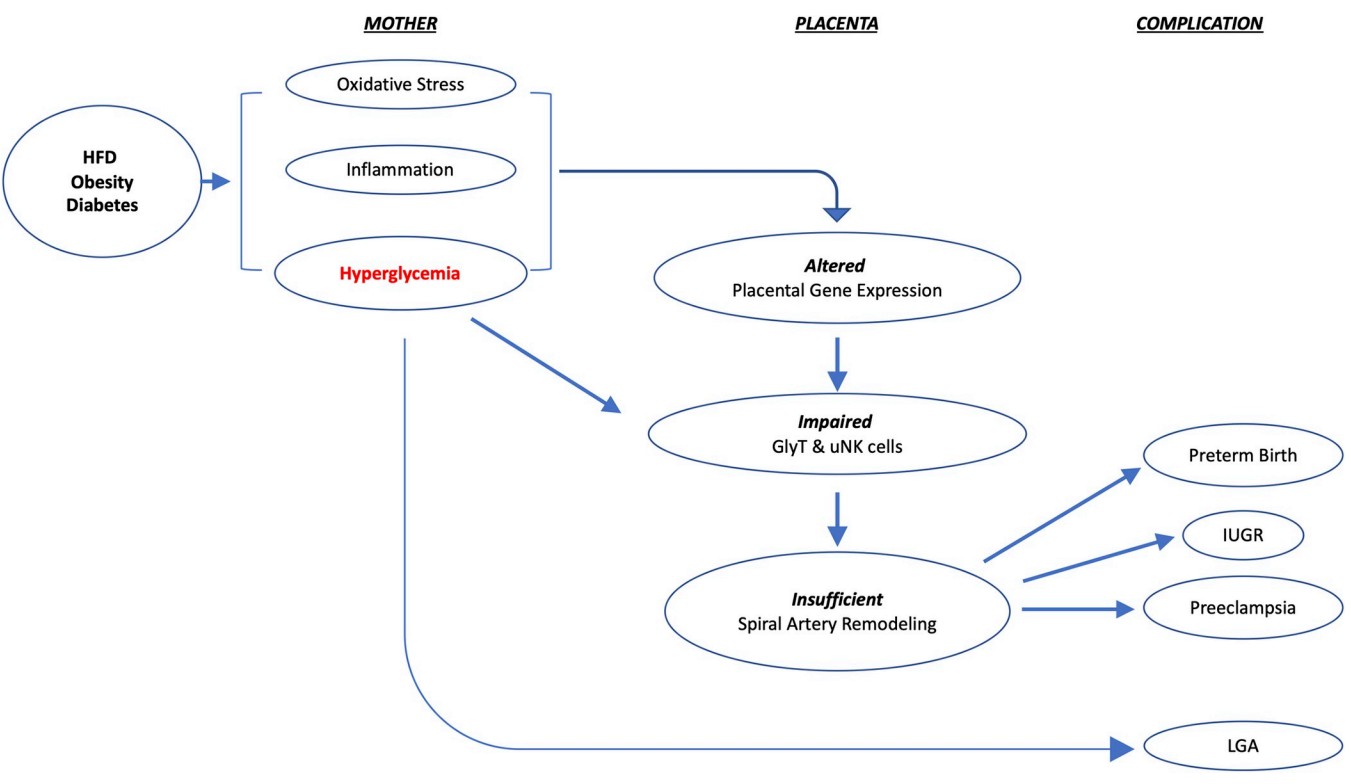

**Fig 9. A hypothesis for the pregnancy disorders caused by a HFD, obesity, and/or diabetes.**

placentas, we thus speculate that an altered glycogen content in uNK cells may have an adverse effect on uNK survival, proliferation, and function, such as the insufficient recruitment of EVTs for spiral artery remodeling, which leads to abnormal placental vascular development (Fig 9).

In a human pregnancy, maternal obesity and diabetes may result in the delivery of an infant being either LGA or IUGR and the specific outcome may be dependent on the timing of the onset of hyperglycemia. Two factors may impact subsequent fetal outcome: the nutrients in the maternal circulation and the transporting efficiency in fetal maternal interface. On one hand, because glucose in the maternal circulation can freely cross the placental barrier to the fetus, maternal hyperglycemia later in pregnancy may provide excessive nutrients to the fetus and lead to an LGA infant (Fig 9). On the other hand, early-onset hyperglycemia (such as in diabetes onset prior to pregnancy) has been found to impair intrauterine trophoblast invasion and disrupt trophoblast stem cell differentiation into EVTs in rats [59], which suggest that hyperglycemia occurring early in pregnancy may lead to insufficient spiral artery remodeling, subsequently impairing the placental vascular formation and nutrient transfer to the fetus, thus resulting in an IUGR infant. In our HFD-PreCP mice, even though some defects in the placental vasculature was observed (Fig 7F), fetal weights were not significantly affected (Fig 3B), perhaps the result of elevated blood glucose levels (Fig 3C), or due to the enlargement of the F-Placentas to counteract the insufficient vasculature (Fig 8A and 8C).

Both similarities and differences exist between the mouse and human placentas regarding glycogen storage. Like GlyTs in the junctional zone of the mouse placenta, the basal plate in the human placenta contains EVTs. EVTs proximal to the labyrinth contain low amounts of glycogen, while distal EVTs are vacuolated and glycogen rich, like GlyTs. In addition, glycogen

is also found in cytotrophoblast cells of the human chorionic villi [33], which resemble the islets of pre-GlyTs in the labyrinth region of the mouse placenta (Fig 5G and 5H). The interstitial invasion by GlyTs into the mouse decidua is comparable with EVTs invading into the human decidua. However, invasion of trophoblast into the decidua in mice is much more shallower than that of human EVTs, which invade deep into the first third of the myometrium [60].

There are a few limitations in our study. Our observed fasting glucose levels after consuming RC (258 ± 29 mg/dL) and HFD (331 ± 33 mg/dL) were slightly higher than the normal means of 10.3 ± 0.58 mmol/L (185 ± 10 mg/dL) and 14.4 ± 0.74 mmol/L (259 ± 13 mg/dL), respectively, reported by Nascimento-Sales et al. for non-pregnant FVB/N mice [47]. This may be due to our short fasting time (4–6 vs 8–12 h), source of blood collected (intracardiac puncture vs tail clipping), pregnancy state (pregnant vs non-pregnant, being increased during pregnancy [19], use of younger mice (9 vs 16 weeks), and time of blood collection (at euthanasia by $CO_2$ asphyxiation vs serially). In addition, we did not measure insulin levels or perform glucose tolerance testing, which might have provided insight into the possible development of insulin resistance following the consumption of a HFD, but it has been shown that FVB/N become more glucose intolerant after consuming a HFD [47]. Furthermore, we also did not directly quantify nor compare the absolute number of PAS-positive cells in each diet-specific placenta, and only report visual observations. And finally, we did not subcategorize placental differences based on fetal sex, which may have revealed sex-dependent changes in placental phenotype in association with alterations in fetal growth and substrate supply [11].

Taken together, our data do, however, shed light on the possible effects of an obesogenic diet consumed during pregnancy on placental gene expression and on two important placental glycogen-rich cell populations, GlyTs (EVTs) and uNK cells. Altered placental genes that are related to oxygen sensing and oxidative stress may contribute to the early development of the placental vasculature as we have previously observed in our studies on the role of *Hmox1* in spiral artery remodeling [34] and by others (cited in [61]). We speculate that a regulatory role of glycogen in GlyTs and uNK cells, beyond that of energy storage, might be a putative link between metabolic disorders, such as obesity and diabetes, with pregnancy complications, especially those due to insufficient spiral artery remodeling. This is supported by clinical observations that aberrant placental glycogen storage is associated with preeclampsia and IUGR [62–64]. In addition, glycogen accumulation in the human placenta primarily occurs during the first trimester when spiral artery remodeling occurs, but less frequently in the second or third trimester, when energy is primarily needed to support fetal growth [65]. Overall, our understanding of glycogen regulation and function in placental cells is very limited and therefore more research on glycogen synthesis, deposition, and degradation in GlyTs and uNK cells are necessary to further elucidate the role(s) of placental glycogen in healthy as well as in pathologic pregnancies.

## Supporting information

**S1 Table. Gene-specific primer sequences used in the qPCR panel.**
(DOCX)

**S2 Table. Weights (in mg) of fetuses and F-placentas harvested at E14.5 and E16.5 from mice subjected to the Control, HFD-P, and HFD-PreCP regimens.** Placental efficiency (PE) was then calculated as the ratio of fetal:F-placental weights.
(DOCX)

## Acknowledgments

We like to thank Mrs. Flora Kalish for her help with maintaining our mouse colonies and the Histo-Tec Laboratory for their help in the histological studies.

## Author Contributions

**Conceptualization:** Hui Zhao.

**Data curation:** Hui Zhao, Ronald J. Wong.

**Formal analysis:** Hui Zhao, Ronald J. Wong, David K. Stevenson.

**Funding acquisition:** Ronald J. Wong, David K. Stevenson.

**Investigation:** Hui Zhao, Ronald J. Wong, David K. Stevenson.

**Methodology:** Hui Zhao, Ronald J. Wong, David K. Stevenson.

**Resources:** Ronald J. Wong, David K. Stevenson.

**Supervision:** Hui Zhao, Ronald J. Wong, David K. Stevenson.

**Validation:** Hui Zhao, David K. Stevenson.

**Visualization:** Hui Zhao, Ronald J. Wong, David K. Stevenson.

**Writing – original draft:** Hui Zhao, Ronald J. Wong, David K. Stevenson.

**Writing – review & editing:** Hui Zhao, Ronald J. Wong, David K. Stevenson.

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
