## [Decision Letter · Decision Letter 0]

4 Oct 2023

PONE-D-23-23164The placental vasculature is affected by changes in gene expression and glycogen-rich cells in a diet-induced obesity mouse modelPLOS ONE

Dear Dr. Wong,

Thank you for submitting your manuscript to PLOS ONE. After careful consideration, we feel that it has merit but does not fully meet PLOS ONE’s publication criteria as it currently stands. Therefore, we invite you to submit a revised version of the manuscript that addresses the points raised during the review process.

We look forward to receiving your revised manuscript.

Kind regards,

Giovanni Tossetta, Ph.D

Academic Editor

PLOS ONE

Journal Requirements:

2. To comply with PLOS ONE submissions requirements, in your Methods section, please provide additional information regarding the experiments involving animals and ensure you have included details on methods of anesthesia and/or analgesia, and efforts to alleviate suffering.

4. We note that Figure 2 in your submission contain copyrighted images. All PLOS content is published under the Creative Commons Attribution License (CC BY 4.0), which means that the manuscript, images, and Supporting Information files will be freely available online, and any third party is permitted to access, download, copy, distribute, and use these materials in any way, even commercially, with proper attribution. For more information, see our copyright guidelines: http://journals.plos.org/plosone/s/licenses-and-copyright.

1. You may seek permission from the original copyright holder of Figure 2 to publish the content specifically under the CC BY 4.0 license.

Reviewers' comments:

Reviewer's Responses to Questions

**Comments to the Author**

1. Is the manuscript technically sound, and do the data support the conclusions?

Reviewer #1: Yes

Reviewer #2: Yes

2. Has the statistical analysis been performed appropriately and rigorously? 

Reviewer #1: Yes

Reviewer #2: I Don't Know

3. Have the authors made all data underlying the findings in their manuscript fully available?

Reviewer #1: Yes

Reviewer #2: Yes

4. Is the manuscript presented in an intelligible fashion and written in standard English?

Reviewer #1: Yes

Reviewer #2: Yes

5. Review Comments to the Author

Reviewer #1: Aferre thoroughly reading the paper, twice, I found no reason for demanding changes nor corrections of the text . Perhaps it could be summarised a little bit as it is rather long. This notwithstanding I recommend to publish it in its current form

Reviewer #2: I read this manuscript and found it very interesting especially nowadays with the prevalence of overweight and obesity.

I have some advice regarding citing many statements both in the introduction and in the discussion .

Many statements need appropriate citing.

6. PLOS authors have the option to publish the peer review history of their article (what does this mean?). If published, this will include your full peer review and any attached files.

Reviewer #1: **Yes: **Juan Carlos BELLO MUÑOZ

Reviewer #2: **Yes: **Naser Al-Husban

---

## [Author Response · Author response to Decision Letter 0]

7 Oct 2023

RESPONSE TO REVIEWERS

REVIEWER 1:

After thoroughly reading the paper, twice, I found no reason for demanding changes nor corrections of the text. Perhaps it could be summarised a little bit as it is rather long. This notwithstanding I recommend to publish it in its current form 

We thank the Reviewer for the positive comments!

REVIEWER 2:

I read this manuscript and found it very interesting especially nowadays with the prevalence of overweight and obesity. I have some advice regarding citing many statements both in the introduction and in the discussion. Many statements need appropriate citing. 

We thank the Reviewer for the positive review. We have now included 8 additional citations (ref #’s 5, 6, 7, 8, 23, 44, 53, and 60) and cited #’s 45, 46 [old #’s 40 and 41] for another statement) in the Introduction and Discussion as recommended.

---

## [Editor Report · Decision Letter 1]

27 Oct 2023

The placental vasculature is affected by changes in gene expression and glycogen-rich cells in a diet-induced obesity mouse model

PONE-D-23-23164R1

Dear Dr. Wong,

We’re pleased to inform you that your manuscript has been judged scientifically suitable for publication and will be formally accepted for publication once it meets all outstanding technical requirements.

Kind regards,

Giovanni Tossetta, Ph.D

Academic Editor

PLOS ONE

---

## [Editor Report · Acceptance letter]

3 Nov 2023

PONE-D-23-23164R1 

The placental vasculature is affected by changes in gene expression and glycogen-rich cells in a diet-induced obesity mouse model 

Dear Dr. Wong:

I'm pleased to inform you that your manuscript has been deemed suitable for publication in PLOS ONE. Congratulations! Your manuscript is now with our production department. 

Kind regards, 

on behalf of

Dr. Giovanni Tossetta 

Academic Editor

PLOS ONE